# Inference Time Alignment for Code Auditing

## Abstract

Software auditing is an increasingly critical task in the era of rapid code generation. While LLM-based auditors have demonstrated strong potential, their effectiveness remains limited by misalignment with the highly complex, domain-specific nature of bug detection. In this work, we introduce BugScope, an inference-time alignment framework inspired by human auditing practices. BugScope structures auditing into three steps: seed identification, context retrieval, and bug detection, and aligns LLMs to each step by analyzing real bug reports, generating diverse examples, and distilling concise, reusable guidelines. On a curated dataset of 40 real-world bugs from 21 widely used open-source projects, BugScope achieves 86.05% precision and 87.88% recall, corresponding to an F1 score of 0.87. By comparison, leading industrial tools such as Cursor BugBot and CodeRabbit achieve F1 scores of only 0.43 and 0.29, respectively. Beyond benchmarks, large-scale evaluation on real-world projects such as the Linux kernel uncovered 184 previously unknown bugs, of which 78 have already been fixed and 7 explicitly confirmed by developers.

## 1 Introduction

Bugs pose a serious threat to software systems. With recent advances in GenAI-driven code generation, code is now being produced at an unprecedented pace, which makes code auditing, a long-standing step in the software development cycle, more critical than ever. Traditionally, auditing has relied on static and symbolic program analysis tools such as Infer (Meta, 2025) and CodeQL (GitHub, 2025). These tools are effective for detecting certain classes of bugs, but they often fall short when deeper semantic reasoning is required, as they typically abstract programs into graphs and perform purely structural reasoning while ignoring symbol-level semantics. LLM-based auditors, such as BugBot (Cursor, 2025) and CodeRabbit (2025), show significant promise since large reasoning models can capture and reason about semantic information more deeply. However, they remain constrained by limited context lengths and a tendency to hallucinate in the presence of complex program structures. To address these challenges, neuro-symbolic approaches (Li et al., 2025c; Guo et al., 2025) integrate traditional program analyses with LLMs, leveraging the strengths of both. Despite encouraging results, these approaches still struggle to balance utility and effectiveness, often leading either to excessive false positives or to overlooked true bugs.

The root cause of the difficulty in using LLMs for code auditing is that they are not inherently aligned with this complex, domain-specific task. Lately, there has been substantial progress in model alignment, which refers to adapting LLMs so that their behavior better matches human goals and domain requirements. For example, alignment through instruction-tuning and reinforcement learning with human feedback (RLHF) (Ouyang et al., 2022) has made models more responsive to natural language instructions, while adversarial alignment techniques (Zou et al., 2024) have improved robustness against malicious inputs. These advances are exemplified by the success of ChatGPT and Claude, which, through large-scale alignment pipelines (Guan et al., 2024; Bai et al., 2022), have substantially improved resilience against prompt injections, jailbreaks, and other security attacks. In particular, *inference-time alignment* techniques (Wang et al., 2024c), where a model is guided or constrained at inference rather than retrained, have shown strong promise. Such methods use structured prompting, external reasoning modules, or rule-based validators to steer model outputs dynamically, reducing hallucinations and improving reliability without costly retraining.

Inspired by these successes, we propose a new inference-time alignment technique that aligns LLMs with the specialized task of software auditing. By integrating structured workflows, domain-specific

constraints, and synthesized guidelines, our approach enables the model to reason about code semantics systematically and reliably, reducing false positives while improving bug detection effectiveness. Specifically, we begin by designing a workflow that mirrors how human experts conduct code audits. Instead of attempting to reason about an entire codebase holistically, auditors follow a disciplined workflow of seed identification, context retrieval, and bug detection. By decomposing the complex task of code auditing into these three stages, alignment becomes a matter of aligning the model to each step individually, thereby reducing task complexity and improving reliability.

In the *learning stage*, we provide the model with representative bug cases and guide it to generate additional and complementary examples along each step of this workflow. These examples are further abstracted to operation guidelines for inference-time alignment. For seed identification, the model enumerates typical program constructs that instantiate potential bug locations, e.g., division operator for divide-by-zero (DBZ) bugs. For the retrieval step, the model explores the different ways of gathering relevant code, such as following *data dependences* to variable definitions. To make retrieval cost-effective, we introduce *termination conditions*, which specify when further expansion of the code context is unnecessary. In the detection step, the model explores *sufficient conditions for bugs*, concise code patterns that enable an auditor to conclude with high confidence that a bug is present. We refer to these patterns as *anti-patterns*. For example, anti-patterns for DBZ include the explicit assignment of zero to a variable or the loading of an external value into a variable, where the latter may evaluate to zero depending on the input. By codifying such anti-patterns, we align the model's focus and reasoning discipline with the requirements of real-world bug detection.

From the generated examples, BUGSCOPE synthesizes two sets of guidelines: (1) a *retrieval strategy*, consisting of a seed extractor and traversal rules for a *context retrieval agent*, and (2) a *detection prompt*, consisting of anti-pattern specifications for a *bug detection agent*. In the subsequent *auditing stage*, these agents collaborate to analyze new projects: the retrieval agent identifies seeds and constructs candidate contexts using the retrieval strategy, while the detection agent applies the detection prompt to those contexts to generate bug reports.

**Results.** We evaluate BUGSCOPE using three recent reasoning models, namely Claude 3.7 Sonnet Thinking, OpenAI o4-mini, and DeepSeek-R1. Experimental results demonstrate that BUGSCOPE consistently outperforms the state-of-the-art LLM-driven code auditor RepoAudit as well as commercial static analysis tools, including Meta Infer, Cursor Bugbot, and CodeRabbit. When powered by Claude 3.7 Sonnet Thinking, BUGSCOPE achieves 88.37% precision, 87.88% recall, and an F1 score of 0.88 in a controlled experiment. In contrast, the highest F1 score among the baselines is only 0.39. For specific anti-patterns, such as negative offset and insufficient zero check, the baselines detect at most one true positive and several of them even fail to identify any buggy cases at all. When deployed on large-scale, real-world codebases, BUGSCOPE uncovers 184 previously unknown bugs, 78 of which have already been fixed and 7 confirmed by developers. Notably, twelve bugs in the Linux kernel discovered by BUGSCOPE have been either fixed or acknowledged by maintainers. These results highlight the practical effectiveness and wide applicability of BUGSCOPE in real-world software auditing. We will publish the bug list upon paper acceptance.

## 2 MOTIVATION

Bug detection is an extremely complex domain-specific task. In this section, we use two real examples to illustrate such complexity and the entailed challenges for LLMs and agents. The code snippets have been substantially simplified for illustration. Figure 1(a) illustrates an out-of-bound (OOB) bug. The original buggy code at line 2 declares a local buffer of size 256, which is passed to function `parse_rtattr()` at line 5. However, inside the function (line 10), `memset()` is invoked with the size of 257. The two functions involved reside in different source files. Figure 1(b) presents a divide-by-zero (DBZ) bug. The variable `u` is obtained through complex computation and may yield a zero value. It is later used as a divisor at line 10.

**Limitations of Existing Techniques** Although these two are of typical bug types: OOB and DBZ, traditional static analysis tools (Sui & Xue, 2016; Arzt et al., 2014; GitHub, 2025; Meta, 2025) have difficulty detecting them due to the tools' inherent limitations in reasoning about deep semantics. In particular, these tools usually ignore semantics in symbols, model programs as graphs, and leverage iterative graph algorithms to derive analysis results. As such, tool developers have to explicitly model a lot of semantics (e.g., manually constructing symbolic summaries of library functions),

(a) OOB bug example in frr

(b) DBZ bug example in openssl

(c) Comparison of basic prompt and prompt synthesized by BUGSCOPE

Figure 1: Motivating examples

which may not be comprehensive to cover a wide spectrum of projects. For example, without the code of `memset()`, which is the typical case as it belongs to a library, or the manually created model for the function, classic tools cannot determine there is an OOB bug in the first case. Without the ability to reasoning about the mathematical semantics of the computation of u in the second case, the DBZ bug is missed as there is not a single zero value assignment of u.

Second, although LLMs have demonstrated the ability to reason about deep semantics (e.g., in pointer aliasing (Guo et al., 2025)), they fall short in detecting these bugs. As shown in Figure 1(c) step ❶, simply providing the bug definition and asking the LLM (Claude 3.7 in our case) to inspect the given code snippet, like in existing code audit agents such as Cursor Bugbot and CodeRabbit, can hardly work. It faces multiple challenges: *(1) retrieving sufficient code fragments in the project for bug detection, and (2) determining true bugs from numerous candidates.* As shown in the conversation box ❶, the LLM reports 11 DBZ for the code snippet in (b) with 10 false positives as it considers all divisions as potentially buggy when it cannot find a zero-value check in the surrounding code region. If we change the prompt to forbid the LLM to report any DBZ bugs unless it has explicit evidence that the divisor may be zero, it reports no bug. LLM has similar problems with the OOB example. It either reports a lot of false positives (e.g., regarding pointer rta at line 8) or misses the real bug. Particularly, it is extremely difficult for the LLM or agent to retrieve the exactly needed contexts for the numerous pointers and buffers in the code snippet for detection.

Third, recent research explores integrating LLMs with traditional static analysis (Wang et al., 2024a; Guo et al., 2025; Naik et al., 2021b; Li et al., 2025b; Yang et al., 2025b) to mitigate the above problems, e.g., by using low-level data-flow to retrieve relevant code and enforcing symbolic rules for bug detection. However, such retrieval often has a hard bound (such as three layers of function calls) to control cost and the rules are often too restrictive due to their symbolic nature. For example, RepoAudit uses a retrieval bound of 3 and only considers DBZ that initiates from a 0 value assignment. As a result, it misses the two example bugs.

> The core limitation is that LLMs are not inherently aligned with complex, domain-specific tasks such as bug detection. As with other AI systems lacking proper alignment, the model may become overly restrictive, reducing utility by missing real bugs, or overly permissive, diminishing effectiveness by generating excessive false positives.

**Our Idea.** Inspired by the recent success in model alignment, such as Constitutional AI (Bai et al., 2022) and Deliberative Alignment (DA) (Guan et al., 2024), which provide domain-specific guidelines that regulate models' reasoning in complex tasks, we propose a novel inference-time alignment technique for code auditing. The essence of recent alignment techniques is to go beyond just telling the model "*what* the task is" (e.g., providing bug definition in our context) and provide reasoning chains on "*how* to complete the task". For instance, a guideline for model safety in DA is: *"Before ..., the model should first recall the safety specification, then reason explicitly about why the request violates it, and only then generate a compliant refusal response."* The guideline forces the model to perform safety reasoning instead of responding without deliberative thinking. Recent research has demonstrated that such reasoning guidelines provide very effective alignment in complex tasks.

A major challenge in bug detection alignment is determining how to provide effective guidance without relying on large sets of manually crafted rules. Real-world software bugs exhibit substantial diversity: the Common Weakness Enumeration (CWE) catalogs more than 900 distinct weakness types (MITRE, 2025a), and continuous reports from platforms such as OSS-Fuzz (Google, 2025) and the CVE (MITRE, 2025b) show that modern systems continue to expose numerous additional, system-specific vulnerabilities. As a consequence, predefined bug patterns are often incomplete, and designing specific detection logic for each bug type remains a time-consuming process.

To cope with this diversity, BUGSCOPE adopts the same discipline used by human auditors: reasoning through a step-by-step process and progressively gathering only the code relevant to a suspected issue. Unlike the holistic approach adopted by most existing LLM-based review agents (e.g., Bugbot, CodeRabbit), where the model is given an entire code segment and asked to uncover all possible bugs, our workflow narrows the model's attention to a focused set of closely related bug instances (e.g., all memory-safety issues involving a particular buffer). This targeted strategy both reduces hallucinations and improves reliability.

In addition to simplifying the workflow to ease alignment, we present challenging examples, i.e., real bug cases with complex causal chains, and prompt a strong reasoning model to reflect on them. The model engages in deeper analysis and generalization to surface relevant prior knowledge (e.g., alternative code patterns that imply similar bugs) and then distills guidelines for each stage of the workflow. This mirrors how human experts calibrate their alertness and rigor by studying historical bug reports before conducting real audits.

Considering DBZ bugs, BUGSCOPE synthesizes a number of guidelines from the example. The highlighted texts in conversations ❷ and ❸ show the retrieval and detection guidelines. Particularly, the retrieval guideline indicates that the agent needs to bring in all the code covering the entire transitive computation of u's value. The bug detection guideline indicates that if 0 is a boundary condition for a variable, the variable may hold a zero value although there is no explicit zero value assignment to the variable. This guideline resembles how a human expert leverages indirect hints during auditing. These guidelines are extremely effective. They are used to detect DBZ in 12 *other* projects and report 23 true bugs with only 5 false positives. Similar results for other bug types are reported in Section 4.2. We will explain more details in the following sections.

## 3  APPROACH

Figure 2(a) illustrates the overview of BUGSCOPE. It consists of two phases as separated by the dashed line: a *learning phase* that automatically derives a set of guidelines for code retrieval and bug detection from a given bug report and an *auditing stage* in which the guidelines are used to align two interacting agents for bug detection. In the learning stage, BUGSCOPE takes an existing bug report as input, extracts descriptive details and relevant code fragments, and guides the LLM to reason through the structured three-step workflow mentioned in the last section. The LLM is further prompted to produce both positive and negative examples illustrating complementary cases across these steps. From these examples, the system abstracts (1) a *retrieval strategy*, consisting of a seed

Figure 2: The workflow of BugScope

extractor and retrieval rules for the *context retrieval agent*, and (2) a *detection prompt*, consisting of detection guidelines for the *bug detection agent*.

In the auditing stage, given a repository, the retrieval agent identifies potential seeds and gathers associated code fragments to construct candidate contexts in line with the learned guidelines. The detection agent then applies the synthesized detection prompt to these contexts to detect bugs. The following subsections describe the design of each stage in detail.

## 3.1 LEARNING STAGE

The goal of the learning stage is to derive the principles/guidelines for detecting a particular kind of bugs. It comprises four steps: *information extraction*, *data augmentation*, *seed extraction and retrieval guideline synthesis* and *detection guideline synthesis*.

**Information Extraction.** A key design decision in BugScope is to initiate guideline derivation from a representative example of a specific bug type. We focus on existing bug reports whose causal chains span multiple functions and whose issue discussions involve more than three rounds of comments. Such cases typically capture richer causal structure and are therefore well-suited for model alignment. However, bug reports are often written in arbitrary formats and may lack critical details (e.g., relevant code snippets or explicit reasoning steps). Consequently, we must extract the necessary information from the associated code repository. As shown in Step (I) of Figure 3, an information-extraction agent processes the bug report and retrieves fix commits, relevant function names, file paths, and pre-/post-fix function bodies through a tool-assisted code lookup mechanism. These steps rely on standard techniques and are not our main contributions; detailed implementation descriptions are provided in Appendix B.

**Data Augmentation.** BugScope begins with a single example for each bug type. While such an example captures some degree of complexity, it is insufficient to represent the full spectrum of possible manifestations. Simply prompting the model to generate more buggy instances of the same type performs poorly, given the vastness of the sample space. Building on the seminal research on *concept learning* (Bruner et al., 1956), which showed that one of the most effective strategies for understanding a complex and unfamiliar domain is to partition it into meaningful dimensions and explore them separately, we adopt a similar strategy. Specifically, we partition the domain of detecting a particular class of bugs into the three dimensions defined by our workflow and then instruct the model to sample along each dimension. The prompt is shown in the "Data Augmentation" conversation in Figure 3, where the three dimensions are highlighted in yellow.

First, the model is prompted to enumerate *seeds* for a given bug type and generate illustrative examples. For instance, seeds for DBZ bugs include both division and modulo operations, while seeds for OOB bugs include buffer indexing and pointer dereferencing after pointer arithmetic. Without considering further complexities, the model can typically enumerate most cases along this dimension. Examples (1) and (2) in Figure 3 (white text boxes) illustrate two seeds for DBZ bugs.

Second, the model is instructed to consider how to retrieve relevant code snippets starting from a seed and when to terminate such retrieval. This process is bug-type specific. For DBZ bugs, relevant code should be collected by tracing *data dependence* and *control dependence* relations in a *backward* fashion. That is, beginning with the seed, one should retrieve variable definitions that contribute, directly or indirectly, to the seed through computations or guarding conditions. In Example (2) of

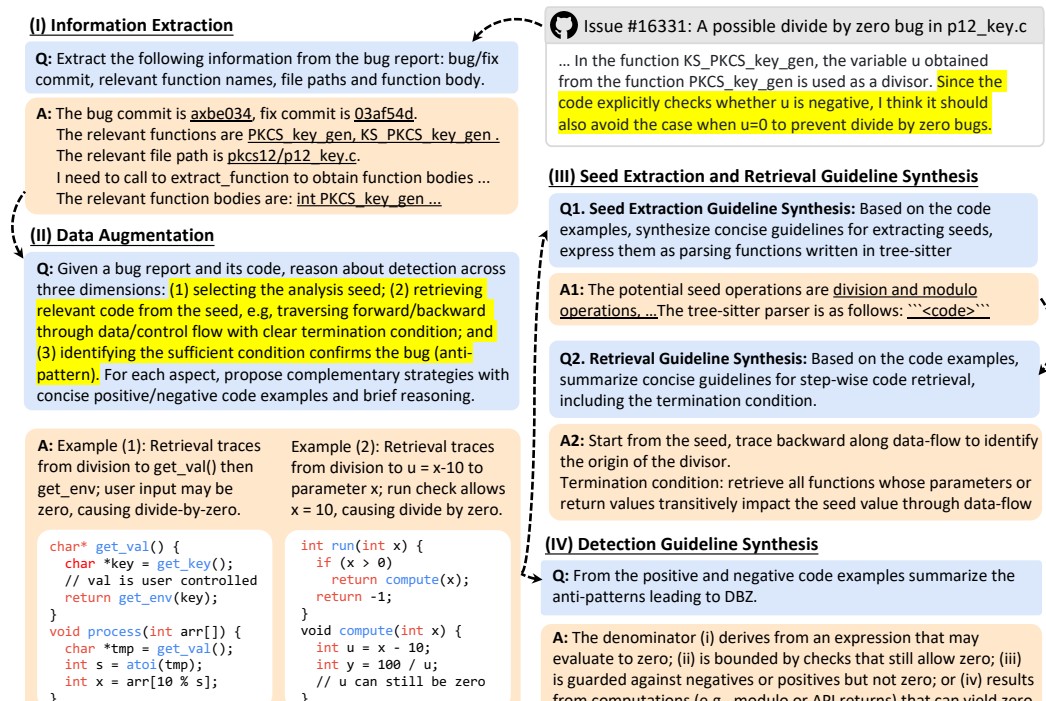

Figure 3: An example of guideline derivation

Figure 3, starting from the division on `u`, the retrieval should transitively include the computation of `u` in the caller function `run()`, especially the guarding condition `x > 0`, which determines whether the division executes. Other retrieval strategies include following *control flow* in a *forward* fashion, as in memory leak (MLK) detection: beginning from an allocation point, one follows program paths forward to check whether the allocated resource is eventually released.

The *termination condition* is essential for cost-effective auditing and varies by bug type too. For DBZ retrieval, a naive strategy is to follow all transitive data dependences until the variable initializations. However, this is often inefficient and unnecessary. In practice, human auditors rely on domain knowledge to terminate earlier. For example, in Example (1) of Figure 3, starting from the seed `10 % s`, the retrieval proceeds to the callee `get_val` because `s` is transitively derived from its return value `tmp`. Inside the callee, the return value is obtained through a library call that reads an environment variable using `key`. At this point, further retrieving the computation of `key` is unnecessary, since the auditor can already conclude that `val` may be zero due to the external call.

Third, the model is instructed to generate examples of possible *sufficient conditions* for a given bug type, namely code patterns that enable an auditor to draw a highly confident conclusion about the presence of a bug. We refer to these patterns as *anti-patterns* in this paper. In practice, when auditing code for a particular class of bugs, human experts rarely rely solely on formal definitions; instead, they actively seek such anti-patterns. For example, an overly restrictive rule used by RepoAudit for DBZ detection involved traversing backward from a seed division operation and checking for a direct zero assignment, thereby avoiding the complexity of reasoning about variable value ranges. In reality, auditors consider a broader set of anti-patterns. One example is the boundary check `u >= 0` in Figure 1(b), which implicitly reveals that 0 is treated as a boundary value, as noted in the corresponding issue report. Another instance, shown in Example (2) of Figure 3, is the guard `x > 0`, which suggests that `x` is positive and may later produce a zero value when evaluated as `x - 10`. A further anti-pattern for OOB appears in `calloc(n, sizeof(T))`, where the computed size `n * sizeof(T)` may overflow and wrap to a smaller value, causing a buffer overflow.

In addition to generating buggy (positive) examples, the LLM is also directed to produce similar but correct (negative) examples, enabling contrastive learning. Each example is paired with a brief explanation. Compared to relying solely on real issue reports, our data augmentation achieves greater diversity by explicitly guiding the model to sample along the three dimensions. Moreover, the generated code examples tend to be concise and focused, avoiding much of the noise present in real-world code, which in turn facilitates more effective downstream guideline synthesis.

**Retrieval Guideline Synthesis.** To synthesize guidelines from the generated examples, we can simply prompt a strong LLM to generate the guidelines directly. However, we found this naive method to be ineffective: the resulting guidelines often had limited coverage and, in some cases, were even incorrect. Instead, our design elicits concrete examples from the model's prior knowledge and then instructs the model to abstract general guidelines from those examples. This is analogous to how legal principles are derived in common law systems (Levi, 1949): courts do not invent rules in the abstract, but rather generalize principles from a body of concrete cases. Through precedent, judges extract recurring patterns and formulate doctrines that guide future decisions.

Technically, a *seed extractor* is synthesized and built upon the parsing library Tree-sitter (Brunsfeld, 2018) for identifying seed locations. For example, a seed extractor for DBZ bugs can be implemented with a simple Tree-sitter query that locates division or modulo expressions, e.g., `binary_expression [operator: "/" or "%"]`. In addition, a set of retrieval guidelines are synthesized to specify how to collect program statements relevant to these seeds and when to stop the retrieval process. The guidelines capture three categories of information: (1) the relations to follow when retrieving additional code, such as function invocations, data dependences, and control dependences; (2) the direction of traversal, forward, backward, or bi-directional; and (3) the stopping conditions. A representative guideline is shown in Step (III) in Figure 3.

Stopping conditions differ by bug type. For DBZ, one synthesized guideline is to terminate at an anti-pattern, since further retrieval is unnecessary once sufficient evidence has been identified. For memory safety bugs such as OOB, one termination condition is to stop at the enclosing function of the buffer variable under inspection. For instance, if a local buffer in function `A()` is passed to `B()`, which then passes it to `C()` where the buffer is accessed, the guideline prescribes backward retrieval starting from the buffer accesses (the seeds) along function calls until reaching the enclosing function `A()`. Together with its callees, `A()` provides the complete context required for OOB auditing.

**Detection Guideline Synthesis.** The goal of this step is to generate specifications of anti-patterns. Step (IV) in Figure 3 illustrates this process with concrete examples for DBZ bug detection. Notably, the resulting guidelines are concise and capture the most common anti-patterns.

### 3.2 AUDITING STAGE

With the guidelines synthesized during the learning stage, BUGSCOPE performs inference-time alignment for code auditing. Given a project and a target bug type, BUGSCOPE first invokes the *context retrieval agent* to identify code fragments that are likely to contain bugs. These candidate fragments are then passed to the *bug detection agent*, which generates bug reports.

The context-retrieval agent follows a neural–symbolic procedure that combines static program structure with LLM-based semantic reasoning. We first parse the repository using Tree-Sitter (Brunsfeld, 2018) to construct abstract syntax trees (ASTs) and an inter-procedural call graph. Starting from each seed statement, the agent performs *semantic slicing*: given a function and a target statement, extracts relevant statements through data and control dependencies. The agent then performs selective inter-procedural expansion, traversing the call graph only along relevant dependency edges and repeating the slicing procedure in functions whose caller–callee relationships are semantically connected to the current slice. This iterative expansion terminates when further traversal is unnecessary for the property being checked. To reduce hallucination during downstream detection, all retrieved statements are aggregated and inlined into a single synthetic function that respects caller–callee ordering. This consolidated context serves as the final output of the context-retrieval agent.

Finally, the retrieved code snippets are embedded into a bug detection prompt that incorporates the detection guidelines. To further reduce false positives, we integrate a set of validators inspired by (Wang et al., 2024b), which check the validity of bug causal chains (e.g., path feasibility). As these mechanisms are not our primary contributions, their details are omitted.

## 4 EVALUATION

We use the Tree-sitter parsing library (Brunsfeld, 2018) to support the context retrieval agent, enabling accurate and scalable syntactic analysis. We evaluate BUGSCOPE on three state-of-the-art models: Claude 3.7 Sonnet Thinking (hereafter Claude 3.7), OpenAI o4-mini, and DeepSeek-R1. We configure Claude 3.7 with a 4,096-token output limit and a 2,048-token reasoning limit, and

Table 1: Statistics of BUGSCOPE with Claude 3.7 Sonnet Thinking. **AP**: Anti-Pattern (Appendix A). **#C**: number of cases. **#Seed**: number of extracted seeds. **#R**: reproduced bugs. **#N**: new bugs. **P(%)**, **R(%)**, **F1**: precision, recall, F1 score. **Time**, **Cost**: total analysis time and financial cost.

| Type | AP | #C | #Seed | #R | #N | #TP | #FP | P(%) | R(%) | F1 | Time (s) | Cost ($) |
|------|-----|----|-------|----|----|-----|-----|-------|--------|------|----------|----------|
| OOB | OSO | 4 | 269 | 3 | 1 | 4 | 2 | 66.67 | 75.00 | 0.71 | 5,547 | 32.73 |
|      | NOF | 4 | 161 | 4 | 0 | 4 | 2 | 66.67 | 100.00 | 0.80 | 2,567 | 12.35 |
|      | ASO | 4 | 27 | 4 | 0 | 4 | 0 | 100.00 | 100.00 | 1.00 | 1,700 | 4.78 |
| DBZ | IZC | 8 | 32 | 5 | 1 | 6 | 1 | 85.71 | 62.50 | 0.72 | 1,548 | 4.24 |
|      | LZD | 4 | 16 | 4 | 5 | 9 | 0 | 100.00 | 100.00 | 1.00 | 653 | 1.19 |
| MLK | UEC | 4 | 31 | 4 | 1 | 5 | 1 | 83.33 | 100.00 | 0.91 | 1,523 | 5.83 |
|      | MSC | 5 | 10 | 5 | 0 | 5 | 0 | 100.00 | 100.00 | 1.00 | 1,910 | 2.24 |
| Total | | 33 | 546 | 29 | 8 | 37 | 6 | **86.05** | **87.88** | **0.87** | 15,448 | 63.36 |

DeepSeek-R1 with a 4,096-token output limit. Since OpenAI o4-mini does not support customization, we adopt its default configuration.

### 4.1 DATASET

We first survey recent works published in top venues in computer security and software engineering, and manually collect the bug reports released by the authors as our dataset (Guo et al., 2022; 2024; Huang et al., 2024; Guo et al., 2025; Shi et al., 2021; 2018) for controlled experiments. As shown in Table 3, the dataset covers three categories of bugs, namely Out-of-Bounds (OOB), Divide-by-Zero (DBZ), and Memory Leak (MLK), which span a wide spectrum of program properties including pointer-related, numeric, and combined properties. Overall, the dataset consists of 40 bugs from 21 open-source projects, which have on average 29K GitHub stars. Each case records the bug-inducing commit and precise location, and we evaluate on the project versions immediately preceding the fixes. For each bug type, we select 2–3 issues for the learning phase and merge the guidelines derived from these cases. In total, 7 bugs are used for learning, while the remaining 33 bugs are reserved for evaluation. Additional details are in Appendix A.

### 4.2 RESULTS

**Performance of BUGSCOPE.** We evaluate BUGSCOPE on the above dataset. To control the cost of evaluation, we limit the seed extraction to the source files containing the original reported bugs, while code from the entire projects may be retrieved. We measure precision, recall, and F1 score, as well as the time and financial cost of each model. As shown in Table 1, BUGSCOPE powered by Claude 3.7 reproduces 29 out of 33 bugs, discovers 8 additional ones, and achieves 86.05% precision, 87.88% recall, and an overall F1 score of 0.87. In terms of efficiency, the average detection cost per seed is $0.12. Results with additional models, including DeepSeek-R1 and OpenAI o4-mini, are reported in Appendix C, where we compare accuracy and cost trade-offs across models.

**Comparison with Baselines.** We compare BUGSCOPE against RepoAudit (Guo et al., 2025), Bug-Bot (Cursor, 2025), CodeRabbit (2025), and Infer (Meta, 2025), using the same dataset and evaluation metrics. As summarized in Table 5, BUGSCOPE consistently outperforms all baselines across all categories. Powered by Claude 3.7, BUGSCOPE achieves an F1 score of 0.87. In contrast, RepoAudit reaches only 34.88% precision, 42.42% recall, and an F1 score of 0.38. BugBot and CodeRabbit yield comparable precision but suffer from poor recall, resulting in F1 scores of 0.43 and 0.29, respectively. Infer detects only a single true bug with an F1 score of 0.05.

These results highlight the limitations of existing tools: RepoAudit is constrained by its data-flow paradigm and restrictive rules, BugBot and CodeRabbit lack sufficient context reasoning, and Infer is bound by its fixed checkers and build dependencies. In contrast, BUGSCOPE generalizes across diverse anti-patterns, delivering both high precision and recall along with auditor-style explanations. Detailed per-pattern comparisons are presented in Appendix D, and additional case studies of Bug-Bot and CodeRabbit are included in Appendix H.

**Ablation Studies.** We evaluate two ablations of BUGSCOPE, namely *NoRetrieval* (removing the context retrieval agent) and *NoAlign* (synthesizing prompts directly from bug reports without the data augmentation and abstraction). As shown in Table 6, removing context retrieval reduces recall

Table 2: The statistics of BUGSCOPE upon six real-world projects. **TP** and **FP** indicates the numbers of true positives and false positives, respectively. **Co** and **Fx** denote the numbers of bugs that are confirmed and fixed, respectively. **P(%)** denotes precision.

| Project | OOB | | | | | DBZ | | | | | MLK | | | | |
|---------|-----|-----|-----|-----|-------|-----|-----|-----|-----|-------|-----|-----|-----|-----|-------|
| | TP | FP | Co | Fx | P (%) | TP | FP | Co | Fx | P (%) | TP | FP | Co | Fx | P (%) |
| vim | 4 | 2 | 0 | 4 | 66.67 | 0 | 1 | 0 | 0 | 0.00 | 2 | 1 | 0 | 2 | 66.67 |
| systemd | 0 | 0 | 0 | 0 | 0.00 | 2 | 1 | 0 | 2 | 66.67 | 0 | 2 | 0 | 0 | 0.00 |
| dynamips | 2 | 0 | 0 | 2 | 100.00 | 0 | 0 | 0 | 0 | 0.00 | 21 | 2 | 0 | 19 | 91.30 |
| zstd | 1 | 1 | 0 | 0 | 50.00 | 4 | 1 | 0 | 1 | 80.00 | 18 | 4 | 0 | 18 | 81.82 |
| openldap | 0 | 2 | 0 | 0 | 0.00 | 0 | 1 | 0 | 0 | 0.00 | 18 | 5 | 0 | 17 | 78.26 |
| git | 1 | 0 | 1 | 0 | 100.00 | 2 | 0 | 2 | 0 | 100.00 | 5 | 1 | 2 | 3 | 83.33 |
| **Total** | 8 | 5 | 1 | 6 | 61.54 | 8 | 4 | 2 | 3 | 66.67 | 64 | 15 | 2 | 59 | 81.01 |

to 42.42% with only 3 new bugs discovered, while removing alignment decreases true positives to 24 and increases false positives to 38, more than six times that of BUGSCOPE. Further analysis and illustrative examples are provided in Appendix E.

**Real-world Impact.** To evaluate the generality of the retrieval strategies and detection prompts synthesized from the seven learning-phase cases, we apply them to six high-profile real-world projects (up to 1M LoC, avg. 654K). To manage overhead, we sample at most 100 seeds per bug type per repository. As shown in Table 2, BUGSCOPE detects 8 DBZ, 8 OOB, and 64 MLK bugs with 76.92% precision, substantially outperformed all baselines. More details are provided in Appendix F.

**More Bug Types.** To evaluate BugScope's performance across a broader range of bug types, we align BUGSCOPE on all seven Linux kernel issues used in prior work (Chen et al., 2025), each representing a system-specific anti-pattern, and use the aligned agents to guide new retrieval strategies and detection prompts. On Linux, BUGSCOPE generalizes system-specific patterns and finds 102 bugs with 91.07% precision. More details are provided in Appendix G.

In total, BUGSCOPE uncovers 182 previously unknown bugs with 84.26% precision, 78 already fixed and 7 confirmed by developers. These results highlight its practical impact in detecting diverse real-world vulnerabilities.

## 5 RELATED WORK

**LLM-driven Bug Detection.** Recent studies leverage LLMs for bug detection through two paradigms: enhancing traditional analyzers or building autonomous agents. In the former, LLMs are used to generate queries or specifications that augment static analysis pipelines (Li et al., 2025c; 2024). In the latter, LLMs act as the primary detection engine, employing techniques such as iterative prompting, subtask decomposition, or cross-referencing against formal specifications (Guo et al., 2025; Wang et al., 2024a; Zheng et al., 2025). While effective, these approaches often target narrow bug classes or require substantial domain expertise. In contrast, our work follows a *learn-from-example* paradigm, enabling LLMs to infer anti-patterns from code examples and achieve broad coverage without specialized analysis infrastructure.

**Customizable Static Analysis.** Static analyzers such as FlowDroid (Arzt et al., 2014), SVF (Sui & Xue, 2016), Infer (Calcagno et al., 2009), and KLEE (Cadar et al., 2008) are typically tailored to specific bug classes, and adapting them requires significant effort. CodeQL (GitHub, 2025) supports customization through user-defined queries but poses a steep learning curve. Recent work has explored automatic checker synthesis from templates or examples (Li et al., 2025a; Yang et al., 2025a; Naik et al., 2021a), yet these methods remain tied to symbolic analysis frameworks. Our approach differs by leveraging LLM generalization to synthesize detection prompts that mimic human reasoning, offering a more flexible and broadly applicable paradigm for bug detection.

## 6 CONCLUSION

This paper introduces BUGSCOPE, an LLM-based multi-agent framework that emulates the human auditing process to learn from examples and detect software bugs. The approach significantly outperforms existing methods and demonstrates strong effectiveness in real-world evaluations.

## 7 ETHICS STATMENT

This work adheres to the ICLR Code of Ethics and addresses several ethical considerations relevant to our research. All issues detected during evaluation are manually validated by the authors before any external communication. Each bug report is independently reviewed by two authors, and disagreements are adjudicated by a third author. Only cases for which all three authors reach consensus are labeled as true bugs and subsequently reported. Importantly, we do not rely on open-source developers to determine whether a detected issue is a true or false positive. Developers are contacted only after the authors have fully validated each bug. As a result, developers are not used as annotators, and no part of our evaluation depends on developer feedback. This avoids ethical risks associated with outsourcing validation to open-source maintainers or involving them in unconsented human-subject roles.

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

# A  DETAILS OF THE DATASET

**Dataset Collection**  To ensure correctness and high-quality ground truth, we collected all bugs from prior works that include manually validated root causes, including 14 Divide-by-Zero cases from (Guo et al., 2022), 15 Out-of-Bounds cases from (Guo et al., 2024), and 11 Memory Leak cases from (Guo et al., 2025). For each case, we manually examined the corresponding issue discussions and code history and labeled the exact root-cause file and context. This careful validation ensures that every case has a reliable, unambiguous root-cause annotation, which is essential for evaluating the quality of end-to-end bug reports. Table 3 presents the detailed statistics of our dataset. For projects with particularly large codebases, we select a relevant subdirectory that encompasses the complete bug trace as the analysis target. The dataset comprises 40 cases drawn from 21 open-source projects, averaging 81K lines of code and 29K GitHub stars. Each case is annotated with the target file containing the bug and linked to the original bug report, facilitating precise localization and reliable evaluation.

For each bug type, we select two to three issues for the learning phase and merge the guidelines derived from these cases to support code auditing. Since each bug type encompasses multiple anti-patterns, we further categorize the cases according to the guidelines synthesized in the learning stage. This categorization allows us to evaluate the performance of different detection tools on specific anti-patterns.

Table 3: The details of the evaluation dataset. Projects marked with * are used for learning and excluded from evaluation.

| Project | Bug Type | Anti-pattern | Commit | Target File | Report |
|---|---|---|---|---|---|
| frr * | OOB | OSO | 26b2fbf | zebra/kernel_netlink.c | link |
| zstd * | OOB | NOF | e5db7c9 | programs/util.c | link |
| systemd * | OOB | ASO | fa2ba7a | src/libsystemd-network/disc-router.c | link |
| redis | OOB | OSO | 93dda65 | src/t_zset.c | link |
| zstd | OOB | OSO | e5db7c9 | programs/util.c | link |
| systemd | OOB | OSO | fa2ba7a | src/basic/time-util.c | link |
| qemu | OOB | OSO | 232e925 | contrib/elf2dmp/qemu_elf.c | link |
| curl | OOB | NOF | e0c68f02 | lib/sendf.c | link |
| curl | OOB | NOF | e0c68f02 | lib/sendf.c | link |
| php-src | OOB | NOF | 492f9c6 | ext/opcache/zend_accelerator_blacklist.c | link |
| openssl | OOB | NOF | 2837b19 | crypto/bf/bf_ofb64.c | link |
| php-src | OOB | ASO | 492f9c6 | sapi/cli/php_cli_server.c | link |
| frr | OOB | ASO | 26b2fbf | bfdd/control.c | link |
| binutils-gdb | OOB | ASO | 4bb461e | ld/libdep_plugin.c | link |
| gcc | OOB | ASO | 9715f10 | libcpp/files.cc | link |
| linux * | DBZ | IZC | 9e9b451 | block/blk-mq-cpumap.c | link |
| linux * | DBZ | LZD | 9e9b451 | drivers/video/logo/pnmtologo.c | link |
| git | DBZ | IZC | 49ac1d3 | git/builtin/pack-objects.c | link |
| binutils-gdb | DBZ | IZC | 2005aa02 | gdb/amd64-tdep.c | link |
| openssl | DBZ | IZC | bc8c3627 | crypto/pkcs12/p12_key.c | link |
| vim | DBZ | IZC | ccfb7c67 | vim/src/misc2.c | link |
| systemd | DBZ | IZC | f6e40037 | src/shared/creds-util.c | link |
| ImageMagick | DBZ | IZC | 442c87b9 | MagickCore/cache.c | link |
| ImageMagick | DBZ | IZC | 442c87b9 | MagickCore/cache.c | link |
| libuv | DBZ | IZC | af1a79cf | src/unix/linux-core.c | link |
| linux | DBZ | LZD | 9e9b451 | lib/math/rational.c | link |
| linux | DBZ | LZD | 9e9b451 | drivers/char/agp/isoch.c | link |
| goaccess | DBZ | LZD | 0abddd5f | src/gholder.c | link |
| goaccess | DBZ | LZD | 0abddd5f | src/gholder.c | link |
| memcached * | MLK | UEC | 6fb5ef7 | restart.c | link |
| libuv * | MLK | MSC | 98a4bab | docs/code/plugin/main.c | link |
| memcached | MLK | UEC | e15e1d6 | memcached.c | link |
| libsass | MLK | UEC | 4da7c4b | src/permutate.hpp | link |
| h3 | MLK | UEC | 3a02395 | src/apps/filters/h3.c | link |
| TrinityEmulator | MLK | UEC | ad25460 | contrib/elf2dmp/main.c | link |
| rtl_433 | MLK | MSC | 474feb5 | rtl_433/src/sdr.c | link |
| linux | MLK | MSC | 4cd8371 | drivers/net/nfpcore/nfp_cppcore.c | link |
| linux | MLK | MSC | 73b73bac | mm/damon/reclaim.c | link |
| TrinityEmulator | MLK | MSC | ad25460 | contrib/elf2dmp/main.c | link |
| binutils-gdb | MLK | MSC | 0ebc886 | binutils/bucomm.c | link |

- **Out-of-Bounds (OOB).** (1) Oversized offset (OSO), where the offset exceeds the buffer length; (2) Negative offset (NOF), where the offset in a buffer operation is negative; (3) Allocation size overflow (ASO), where integer overflow during memory allocation produces a smaller-than-expected buffer.

- **Divide-by-Zero (DBZ).** (1) Insufficient zero check (IZC), where a conditional check on the divisor is insufficient to exclude zero; (2) Literal zero division (LZD), where a literal zero value is directly used as a divisor without validation.

- **Memory Leak (MLK).** (1) Unexecuted cleanup (UEC), where cleanup code exists but is skipped due to early returns or exceptions; (2) Missing cleanup (MSC), where no code is provided to free allocated memory.

**Data Split**  For each bug type, we select 2–3 issues for the learning phase based on the criterion of prioritizing cases whose causal chains span multiple functions and whose issue discussions contain more than three rounds of comments. Such cases offer richer reasoning trajectories for synthesizing robust retrieval strategies and detection guidelines. In total, 7 bugs are used for learning, and the remaining 33 bugs are reserved exclusively for evaluation.

**Discussion on Overfitting Risk**   In our benchmark, 87.88% of the evaluation cases originate from projects that are entirely absent from the learning phase. When excluding these overlapping cases, BUGSCOPE powered by Claude 3.7 achieves 86.49% precision, 86.21% recall, and an F1 score of 0.86. These results are almost identical to the original performance numbers (86.05% precision, 87.88% recall, and 0.87 F1), which indicates that there is no overfitting risk.

## B  INFORMATION EXTRACTION DETAILS

For each issue used in the learning phase, we extract structured information from both the bug report and the corresponding code repository.

**Textual Information.**   We first collect the basic textual components of the issue, including the title, description, developer comments, the bug commit, and the associated fix commit. These elements provide essential high-level context and often contain implicit clues regarding the root cause or affected components. For the bug commit, we first attempt to extract the commit identifier directly from the description; if this information is missing, we approximate it by selecting the commit immediately preceding the issue creation time. This heuristic maximizes the likelihood that the extracted code corresponds to the version referenced in the bug report.

**Semantic Information Extraction.**   The aggregated textual content is then passed to an information-extraction agent, which identifies key semantic elements such as relevant function names, file paths, and bug commits. The agent is prompted to reason about the likely code locations involved in the bug based on the bug description and related contextual information. To retrieve the precise code regions, we equip the agent with a tool named `extract_function`, which locates function bodies using the extracted function name, file path, and commit.

These steps collectively prepare the structured input required for downstream context retrieval and alignment synthesis. While necessary for the overall workflow, they rely on standard techniques and are not considered part of the novel contributions of BUGSCOPE.

## C  EVALUATION WITH MORE REASONING MODELS

To assess the robustness of BUGSCOPE across different reasoning engines, we further evaluate it with DeepSeek-R1 and OpenAI o4-mini in addition to Claude 3.7. The results are summarized in Table 4. When powered by Claude 3.7, BUGSCOPE achieves the best performance, reproducing 29 of 33 bugs, discovering 8 additional ones, and reaching 86.05% precision, 87.88% recall, and an F1 score of 0.87. Both DeepSeek-R1 and o4-mini maintain strong performance, achieving F1 scores of 0.84. In terms of efficiency, the average detection cost per seed is significantly lower with DeepSeek-R1 and o4-mini (about $0.03) than with Claude 3.7 ($0.12). These results suggest that while Claude 3.7 provides the best overall accuracy, DeepSeek-R1 and o4-mini offer more cost-effective alternatives with competitive precision and recall.

## D  DETAILED COMPARISON RESULT WITH BASELINES

**Setup and Metrics.**   To evaluate the effectiveness of BUGSCOPE, we compare it against the state-of-the-art LLM-driven bug detection tool RepoAudit (Guo et al., 2025), as well as three commercial tools, namely BugBot (Cursor, 2025), CodeRabbit (2025), and Infer (Meta, 2025). RepoAudit is an LLM agent that detects data-flow bugs in repositories. BugBot and CodeRabbit are commercial agents that can be integrated with GitHub repositories to automatically scan pull requests. Infer is a production-grade static analyzer with a fixed set of built-in checkers.

We evaluate all tools on the dataset introduced in Section 4.1. For fairness, we only compare results for the cases included in the evaluation set. We use Claude 3.7, the most effective reasoning model (see Section 4.2), to power RepoAudit. We extend RepoAudit with source and sink extractors for

Table 4: Statistics of BUGSCOPE with ✳Claude 3.7 Sonnet Thinking, 🐋DeepSeek-R1, and ⓢOpenAI o4-mini. **AP**: Anti-Pattern. **#C**: number of cases. **#Seed**: number of extracted seeds. **#R**: number of reproduced bugs. **#N**: number of new bugs. **P(%)**, **R(%)**, **F1**: precision, recall, F1 score. **Time**, **Cost**: total analysis time and financial cost.

| Model | Type | AP | #C | #Seed | #R | #N | #TP | #FP | P(%) | R(%) | F1 | Time (s) | Cost ($) |
|---|---|---|---|---|---|---|---|---|---|---|---|---|---|
| ✳ | OOB | OSO | 4 | 269 | 3 | 1 | 4 | 2 | 66.67 | 75.00 | 0.71 | 5,547 | 32.73 |
| | | NOF | 4 | 161 | 4 | 0 | 4 | 2 | 66.67 | 100.00 | 0.80 | 2,567 | 12.35 |
| | | ASO | 4 | 27 | 4 | 0 | 4 | 0 | 100.00 | 100.00 | 1.00 | 1,700 | 4.78 |
| | DBZ | IZC | 8 | 32 | 5 | 1 | 6 | 1 | 85.71 | 62.50 | 0.72 | 1,548 | 4.24 |
| | | LZD | 4 | 16 | 4 | 5 | 9 | 0 | 100.00 | 100.00 | 1.00 | 653 | 1.19 |
| | MLK | UEC | 4 | 31 | 4 | 1 | 5 | 1 | 83.33 | 100.00 | 0.91 | 1,523 | 5.83 |
| | | MSC | 5 | 10 | 5 | 0 | 5 | 0 | 100.00 | 100.00 | 1.00 | 1,910 | 2.24 |
| | Total | | 33 | 546 | 29 | 8 | 37 | 6 | **86.05** | **87.88** | **0.87** | 15,448 | 63.36 |
| 🐋 | OOB | OSO | 4 | 269 | 2 | 1 | 3 | 1 | 75.00 | 50.00 | 0.60 | 12,036 | 6.06 |
| | | NOF | 4 | 161 | 4 | 0 | 4 | 1 | 80.00 | 100.00 | 0.89 | 2,552 | 1.56 |
| | | ASO | 4 | 27 | 4 | 0 | 4 | 0 | 100.00 | 100.00 | 1.00 | 1,362 | 0.80 |
| | DBZ | IZC | 8 | 32 | 4 | 1 | 5 | 0 | 100.00 | 50.00 | 0.67 | 1,548 | 4.24 |
| | | LZD | 4 | 16 | 4 | 3 | 7 | 1 | 87.50 | 100.00 | 0.93 | 653 | 1.19 |
| | MLK | UEC | 4 | 31 | 4 | 1 | 5 | 1 | 83.33 | 100.00 | 0.91 | 1,166 | 0.79 |
| | | MSC | 5 | 10 | 4 | 0 | 4 | 0 | 100.00 | 80.00 | 0.89 | 2,358 | 0.83 |
| | Total | | 33 | 546 | 26 | 6 | 32 | 4 | **88.89** | **78.79** | **0.84** | 21,675 | 15.48 |
| ⓢ | OOB | OSO | 4 | 269 | 3 | 0 | 3 | 2 | 60.00 | 75.00 | 0.67 | 9,161 | 7.83 |
| | | NOF | 4 | 161 | 4 | 0 | 4 | 1 | 80.00 | 100.00 | 0.89 | 1,615 | 2.63 |
| | | ASO | 4 | 27 | 4 | 0 | 4 | 0 | 100.00 | 100.00 | 1.00 | 803 | 1.01 |
| | DBZ | IZC | 8 | 32 | 4 | 0 | 4 | 0 | 100.00 | 50.00 | 0.67 | 1,146 | 1.83 |
| | | LZD | 4 | 16 | 4 | 2 | 6 | 0 | 100.00 | 100.00 | 1.00 | 371 | 0.49 |
| | MLK | UEC | 4 | 31 | 4 | 1 | 5 | 2 | 71.43 | 100.00 | 0.83 | 648 | 0.97 |
| | | MSC | 5 | 10 | 4 | 1 | 5 | 0 | 100.00 | 80.00 | 0.89 | 752 | 0.67 |
| | Total | | 33 | 546 | 27 | 4 | 31 | 5 | **86.11** | **81.82** | **0.84** | 14,496 | 15.43 |

DBZ and OOB and also restrict the scope to the file containing the original bug. For BugBot and CodeRabbit, we simulate GitHub pull requests containing the target file and collect reported issues matching the relevant bug type. Since Infer is compiler-based, we run it from the project root and only consider reports relevant to the target bug type within the corresponding file. We measure precision, recall, F1 score, and the number of newly discovered bugs.

**Results.** As shown in Table 5, BUGSCOPE significantly outperforms all baselines across OOB, DBZ, and MLK categories. RepoAudit achieves 34.88% precision and 42.42% recall, much lower than BUGSCOPE. Although BugBot and CodeRabbit attain comparable precision, their recall remains poor, leading to low F1 scores of 0.43 and 0.29. RepoAudit demonstrates strong recall on MLK bugs but suffers from low precision on DBZ and OOB, yielding an overall F1 of 0.38. Infer performs the worst, detecting only one OOB bug with an F1 of 0.05.

RepoAudit adopts a source–sink paradigm tailored to data-flow vulnerabilities. While this supports customization of sources and sinks, many DBZ and OOB bugs cannot be expressed as data-flow reachability problems, which require reasoning about numerical relationships and multi-domain properties. Consequently, RepoAudit achieves high F1 on unexecuted cleanup (UEC) and missing cleanup (MSC) but fails on patterns such as negative offset (NOF) and insufficient zero check (IZC).

BugBot and CodeRabbit also struggle with complex anti-patterns. They rely on LLM prior knowledge to detect common issues but fail on intricate patterns such as NOF, ASO, and IZC. Moreover, both suffer from insufficient context retrieval. BugBot supports limited inter-procedural reasoning within a single file but cannot trace bug propagation across files, leading to poor coverage and many false negatives. CodeRabbit exhibits even weaker retrieval, and its explanations are overly generic. For example, among its two true positives of literal zero division (LZD), the explanation only states that "if the divisor is zero, a divide-by-zero error may occur," without justifying why the divisor can

be zero. In contrast, BUGSCOPE reconstructs the full propagation path and provides auditor-style explanations. Case studies of BugBot and CodeRabbit are presented in Appendix H.

Infer further suffers from build failures and limited checkers. We were able to evaluate only 22 of 40 cases; others failed due to incompatibilities with Infer's build interception (`infer capture`) under GCC-specific flags, custom Makefiles, or non-standard toolchains. Even among successful builds, Infer detected only one true positive and six false positives. Notably, it has no checker for DBZ, resulting in zero reports for this category. For OOB, although related checkers exist, Infer cannot distinguish between OSO and NOF, limiting its effectiveness.

In contrast, BUGSCOPE replicates the human process of learning anti-patterns. By leveraging the generalization ability of LLM-based reasoning and the synthesized detection prompt, it adapts to diverse bug types and consistently delivers high precision, recall, and informative explanations, even where traditional or commercial tools fail.

**Financial and Computational Cost.** We also compare the monetary and computational cost of BUGSCOPE against baselines. Among the baselines, RepoAudit is the only tool that publicly reports per-query usage cost. Under the same evaluation settings, RepoAudit incurs an average cost of $0.26 per seed, which is higher than the cost of BUGSCOPE. BugBot and CodeRabbit are subscription-based commercial products whose APIs do not expose per-project or per-seed pricing information, making a fine-grained cost comparison infeasible. Infer, in contrast, is a purely symbolic bug detector and therefore incurs negligible monetary cost aside from CPU and memory consumption, but its precision and recall are significantly lower on our benchmark, as shown in Table 5. Overall, BUGSCOPE achieves a favorable balance between effectiveness and cost, delivering substantially higher detection performance while maintaining competitive per-query expenditure.

## E  ABLATION STUDIES

**Setup and Metrics.** We evaluate two ablations, namely *NoRetrieval* and *NoAlign*, to assess the contributions of context retrieval and learning stage alignment, respectively. Specifically, *NoRetrieval* removes the context retrieval agent and provides the bug detection agent only with the file containing the original bug. *NoAlign* synthesizes prompts directly from bug reports without the data augmentation and abstraction. We report precision, recall, and F1 score for both ablations, and also track the number of newly discovered bugs.

**Results.** Table 6 presents the ablation results using Claude 3.7. Without context retrieval, *NoRetrieval* only discovers 3 new bugs and decreases the number of reproduced bugs by 15, reducing the recall to 42.42%. This decline is attributed to the lack of contextual information. In real-world programs, the values that trigger bugs may propagate across multiple functions. For example, in Figure 1(a), both the buffer and the index are passed from the caller function located in another file. Without visibility into the complete data-flow chain, neither human auditors nor the LLM can reliably determine the bug. To address this, we design a context retrieval agent that gathers the relevant program context and, through synthesized retrieval strategies, aligns with the reasoning process of human auditors. This alignment expands the agent's scope and enables more precise bug detection.

When the prompt is synthesized directly from existing bug reports (*NoAlign*), the number of true positives decreases to 24 while false positives increase to 38, more than six times that of BUGSCOPE. This surge in false positives arises from two main factors. First, the functions referenced in bug reports often contain large portions of code unrelated to the target anti-pattern, which distracts the model. Second, without data augmentation, the model-generated retrieval strategy and detection logic tend to overfit the specific code fragments described in the bug report and fail to generalize to other scenarios. For example, in the case of insufficient zero check (IZC), the true bug condition is that the divisor is bounded by a check but zero remains a possible value. Without alignment, however, the synthesized rule collapses into a shallow heuristic such as *check whether the divisor is validated before division*. As a result, the detection agent simply flags any division without an explicit local check, leading to numerous false positives.

Table 5: The comparison results between BUGSCOPE and baselines. **AP** indicates **Anti-Pattern**. **#C**: number of cases. **#R**: reproduced bugs. **#N**: new bugs. **#TP / #FP**: true / false positives. **P(%)**, **R(%)**, **F1**: precision, recall, F1 score.

| Tool Name | Type | AP | #C | #R | #N | #TP | #FP | P(%) | R(%) | F1 |
|---|---|---|---|---|---|---|---|---|---|---|
| **BugScope** | OOB | OSO | 4 | 3 | 1 | 4 | 2 | 66.67 | 75.00 | 0.71 |
| | | NOF | 4 | 4 | 0 | 4 | 2 | 66.67 | 100.00 | 0.80 |
| | | ASO | 4 | 4 | 0 | 4 | 0 | 100.00 | 100.00 | 1.00 |
| | DBZ | IZC | 8 | 5 | 1 | 6 | 1 | 85.71 | 62.50 | 0.72 |
| | | LZD | 4 | 4 | 5 | 9 | 0 | 100.00 | 100.00 | 1.00 |
| | MLK | UEC | 4 | 4 | 1 | 5 | 1 | 83.33 | 100.00 | 0.91 |
| | | MSC | 5 | 5 | 0 | 5 | 0 | 100.00 | 100.00 | 1.00 |
| | Total | | 33 | 29 | 8 | 37 | 6 | **86.05** | **87.88** | **0.87** |
| **RepoAudit** | OOB | OSO | 4 | 1 | 0 | 1 | 4 | 20.00 | 25.00 | 0.22 |
| | | NOF | 4 | 0 | 0 | 0 | 1 | 0.00 | 0.00 | 0.00 |
| | | ASO | 4 | 1 | 0 | 1 | 2 | 33.33 | 25.00 | 0.29 |
| | DBZ | IZC | 8 | 1 | 0 | 1 | 14 | 6.67 | 12.50 | 0.09 |
| | | LZD | 4 | 3 | 0 | 3 | 6 | 33.33 | 75.00 | 0.46 |
| | MLK | UEC | 4 | 4 | 1 | 5 | 1 | 83.33 | 100.00 | 0.91 |
| | | MSC | 5 | 4 | 0 | 4 | 0 | 100.00 | 80.00 | 0.89 |
| | Total | | 33 | 14 | 1 | 15 | 28 | **34.88** | **42.42** | **0.38** |
| **BugBot** | OOB | OSO | 4 | 1 | 1 | 2 | 1 | 66.67 | 25.00 | 0.36 |
| | | NOF | 4 | 1 | 0 | 1 | 2 | 33.33 | 25.00 | 0.29 |
| | | ASO | 4 | 0 | 1 | 1 | 1 | 50.00 | 0.00 | 0.00 |
| | DBZ | IZC | 8 | 1 | 0 | 1 | 0 | 100.00 | 12.50 | 0.22 |
| | | LZD | 4 | 3 | 1 | 4 | 0 | 100.00 | 75.00 | 0.86 |
| | MLK | UEC | 4 | 1 | 1 | 2 | 2 | 50.00 | 25.00 | 0.33 |
| | | MSC | 5 | 3 | 1 | 4 | 0 | 100.00 | 60.00 | 0.75 |
| | Total | | 33 | 10 | 5 | 15 | 6 | **71.43** | **30.30** | **0.43** |
| **CodeRabbit** | OOB | OSO | 4 | 0 | 0 | 0 | 3 | 0.00 | 0.00 | 0.00 |
| | | NOF | 4 | 0 | 0 | 0 | 0 | 0.00 | 0.00 | 0.00 |
| | | ASO | 4 | 1 | 0 | 1 | 0 | 100.00 | 25.00 | 0.40 |
| | DBZ | IZC | 8 | 0 | 0 | 0 | 0 | 0.00 | 0.00 | 0.00 |
| | | LZD | 4 | 1 | 1 | 2 | 0 | 100.00 | 25.00 | 0.40 |
| | MLK | UEC | 4 | 0 | 0 | 0 | 0 | 0.00 | 0.00 | 0.00 |
| | | MSC | 5 | 4 | 0 | 4 | 0 | 100.00 | 80.00 | 0.89 |
| | Total | | 33 | 6 | 1 | 7 | 3 | **70.00** | **18.18** | **0.29** |
| **Infer** | OOB | OSO | 4 | 0 | 0 | 0 | 6 | 0.00 | 0.00 | 0.00 |
| | | NOF | 4 | 1 | 0 | 1 | 1 | 50.00 | 25.00 | 0.33 |
| | | ASO | 4 | 0 | 0 | 0 | 0 | 0.00 | 0.00 | 0.00 |
| | DBZ | IZC | 8 | 0 | 0 | 0 | 0 | 0.00 | 0.00 | 0.00 |
| | | LZD | 4 | 0 | 0 | 0 | 0 | 0.00 | 0.00 | 0.00 |
| | MLK | UEC | 4 | 0 | 0 | 0 | 0 | 0.00 | 0.00 | 0.00 |
| | | MSC | 5 | 0 | 0 | 0 | 0 | 0.00 | 0.00 | 0.00 |
| | Total | | 33 | 1 | 0 | 1 | 7 | **12.50** | **3.03** | **0.05** |

# F EVALUATION ON REAL-WORLD PROJECTS

**Setup and Metrics.** To assess the generality of the retrieval strategies and detection prompts synthesized from the seven learning-phase cases, we selected six high-profile C/C++ projects on GitHub following two criteria: (1) large-scale codebases (over 100K LoC), and (2) active maintenance (substantial development activity within the past six months). These six projects are representative real-world systems, averaging 654K LoC and 22.3K GitHub stars.

For each project, we applied the synthesized retrieval strategies and detection prompts to detect new bugs. To control computational cost, we randomly sampled 100 seed statements per bug type per

Table 6: Results of NoRetrieval and NoAlign. **AP** indicates **Anti-Pattern**. **#C** denotes the number of cases for each anti-pattern. **#R** denotes the number of reproduced bugs. **#N** denotes the number of new bugs found. **#TP** and **#FP** are the number of true and false positives, respectively. **P(%)** and **R(%)** denote precision and recall. **F1** denotes the F1 score.

| Type | AP | #C | NoRetrieval | | | | | | | NoAlign | | | | | | |
|------|-----|----|-----|-----|-----|-----|--------|--------|------|-----|-----|-----|-----|--------|--------|------|
| | | | #R | #N | #TP | #FP | P(%) | R(%) | F1 | #R | #N | #TP | #FP | P(%) | R(%) | F1 |
| OOB | BOF | 4 | 0 | 1 | 1 | 0 | 100.00 | 0.00 | 0.00 | 2 | 1 | 3 | 2 | 60.00 | 50.00 | 0.55 |
| | BUF | 4 | 1 | 0 | 1 | 2 | 33.33 | 25.00 | 0.29 | 4 | 0 | 4 | 7 | 36.36 | 100.00 | 0.53 |
| | ASO | 4 | 3 | 0 | 3 | 1 | 75.00 | 75.00 | 0.75 | 4 | 0 | 4 | 6 | 40.00 | 100.00 | 0.57 |
| DBZ | IZC | 8 | 1 | 0 | 1 | 0 | 100.00 | 12.50 | 0.22 | 2 | 0 | 2 | 5 | 28.57 | 25.00 | 0.27 |
| | LZD | 4 | 3 | 2 | 5 | 1 | 83.33 | 75.00 | 0.79 | 2 | 2 | 4 | 1 | 80.00 | 50.00 | 0.62 |
| MLK | CNE | 4 | 2 | 0 | 2 | 3 | 40.00 | 50.00 | 0.44 | 2 | 0 | 2 | 5 | 28.57 | 50.00 | 0.36 |
| | NCC | 5 | 4 | 0 | 4 | 1 | 80.00 | 80.00 | 0.80 | 5 | 0 | 5 | 2 | 71.43 | 100.00 | 0.83 |
| Total | | 33 | 14 | 3 | 17 | 8 | 68.00 | 42.42 | 0.52 | 21 | 3 | 24 | 28 | 46.15 | 63.64 | 0.54 |

Table 7: The comparison results between BUGSCOPE and baselines on real-world projects. **TP** and **FP** denote the numbers of true and false positives, respectively. **P(%)** denotes precision.

| Tool | OOB | | | DBZ | | | MLK | | | Total | | |
|------|-----|-----|-------|-----|-----|-------|-----|-----|-------|-----|-----|-------|
| | TP | FP | P(%) | TP | FP | P(%) | TP | FP | P(%) | TP | FP | P(%) |
| BugScope | 8 | 5 | 61.54 | 8 | 4 | 66.67 | 64 | 15 | 81.01 | 80 | 24 | 76.92 |
| RepoAudit | 1 | 13 | 7.14 | 0 | 24 | 0.00 | 58 | 14 | 80.56 | 59 | 51 | 53.64 |
| BugBot | 0 | 2 | 0.00 | 0 | 0 | 0.00 | 3 | 1 | 75.00 | 3 | 3 | 50.00 |
| CodeRabbit | 1 | 8 | 11.11 | 0 | 0 | 0.00 | 11 | 5 | 68.75 | 12 | 13 | 48.00 |
| Infer | 0 | 0 | 0.00 | 0 | 0 | 0.00 | 6 | 21 | 22.22 | 6 | 21 | 22.22 |

repository. All baselines were evaluated using the same seed statements and project repositories to ensure a fair comparison. Since ground-truth labels for recall are unavailable in this real-world setting, we report precision only. Each detected bug report was independently examined by two authors, with disagreements adjudicated by a third author. Only cases where all three reviewers reached consensus were considered true positives (TP) and subsequently reported to developers.

**Results.** As shown in Table 2, BUGSCOPE identifies 8 DBZ, 8 OOB, and 64 MLK bugs, yielding an overall precision of 76.92%. Precision is highest for MLK at 81.01%, while DBZ and OOB achieve 66.67% and 61.54% respectively, largely due to the limited backward control-flow and data-flow context available from the extracted seeds.

A comparative analysis with existing baselines is presented in Table 7. RepoAudit performs comparably to BUGSCOPE on MLK bugs, but it identifies only one true bug across OOB and DBZ while producing 37 false positives, resulting in a substantially lower overall precision of 53.64%. BugBot and CodeRabbit detect only 3 and 12 bugs, achieving precisions of 50.00% and 48.00%, respectively. Infer produces no valid reports for OOB or DBZ and detects only 6 MLK bugs, with a precision of 22.22%. Across all six real-world projects, BUGSCOPE consistently surpasses these baselines in both the number of detected issues and overall precision, demonstrating strong robustness and generality across large and diverse codebases.

## G    EVALUATION ON LINUX FOR ADDITIONAL BUG TYPES

**Setup and Metrics.** To evaluate BugScope's performance across a broader range of bug types, we conducted an additional controlled experiment on the Linux kernel. We used all patches from prior work (Chen et al., 2025) as examples for the learning phase, each representing a system-specific anti-pattern, and used the synthesized retrieval strategies and detection logic to identify new bugs in the latest Linux version.

**Results.** As shown in Table 8, we successfully uncovers 102 new bugs in the current Linux kernel with the precision of 91.07%. These bugs span multiple categories, including Use-After-Free (UAF), Wrong Error Code (WEC), UnInitialized Variables (UIV), Null Pointer Dereference (NPD), Divide-

Table 8: Results of BUGSCOPE on Linux. **#TP** and **#FP** are the number of true and false positives, respectively.

| Bug Type | Original Patch | # TP | # FP | Precision (%) |
|---|---|---|---|---|
| Null Pointer Dereference | link | 39 | 3 | 92.86 |
| Divide by Zero | link | 15 | 1 | 93.75 |
| Out of Bounds | link | 7 | 2 | 77.78 |
| Wrong Error Code | link | 20 | 2 | 90.91 |
| Use After Free | link | 1 | 0 | 100.00 |
| Uninitialized Value | link | 12 | 1 | 92.31 |
| Memory Leak | link | 8 | 1 | 88.89 |
| **Total** | — | **102** | **10** | **91.07%** |

Listing 1: An OOB false negative example missed by Cursor BugBot and CodeRabbit.

```
1 static int parse_encap_seg6(struct rtattr *tb, struct in6_addr *segs){
2   struct rtattr *tb_encap[256] = {};
3 netlink_parse_rtattr_nested(tb_encap, 256, tb);
4   ...
5   return 0;
6 }

1 void netlink_parse_rtattr_nested(struct rtattr **tb, int max, struct rtattr *rta){
2   netlink_parse_rtattr(tb, max, RTA_DATA(rta), RTA_PAYLOAD(rta));
3 }

1 void netlink_parse_rtattr(struct rtattr **tb, int max, struct rtattr *rta, int len){
2   memset(tb, 0, sizeof(struct rtattr *) * (max + 1));
3   ...
4 }
```

By-Zero (DBZ), Memory-LeaK (MLK), and Out-Of-Bounds (OOB), demonstrating BugScope's robustness across diverse bug types.

# H  ANALYSIS OF BUGBOT AND CODERABBIT

We evaluate two industrial LLM-based bug detectors, namely Cursor BugBot and CodeRabbit, upon our curated dataset. Both tools are commercial agents that can be integrated within GitHub repositories for automatically scanning pull requests. In our experiments, we simulate a GitHub pull request containing the target file associated with the original bug and collect only those reported issues that match the intended bug type for reproduction. We present detailed evaluation results, including links to the corresponding issues identified by each tool, along with representative examples of false positives and false negatives.

## H.1  EVALUATION RESULT

The detailed evaluation results for BugBot and CodeRabbit are presented in Table 9. Overall, Bug-Bot successfully reproduces 11 known cases and 4 new bugs, including 4 out-of-bounds bugs, 5 divide-by-zero bugs, and 6 memory leak bugs. In comparison, CodeRabbit reproduces 7 cases and discovers 3 new bugs, most of which are memory leak bugs. Notably, CodeRabbit detects only 1 out-of-bounds bug and 2 divide-by-zero bugs, showing its difficulty in handling complex bug patterns. In general, both detectors demonstrate relatively high precision, but suffer from low recall.

## H.2  EXAMPLES OF FALSE NEGATIVES

In Listing 1, the function `parse_encap_seg6` defines an array with 256 elements and passes it to `netlink_parse_rtattr_nested`, along with the parameter `max` set to 256. This function then forwards both arguments to `netlink_parse_rtattr`, which eventually calls `memset(tb, 0, sizeof(struct rtattr *) * (max + 1))`, resulting in an out-of-bounds write. This bug is successfully detected by BUGSCOPE but missed by both BugBot and CodeRabbit. In practice, the functions involved in this case span multiple files, each containing thousands of lines of code. Due to limited context length, both BugBot and CodeRabbit fail to recover the inter-procedural dependencies required to detect this vulnerability.

Table 9: The detailed evaluation results of Cursor Bugbot and CodeRabbit. **BT** indicates **Bug Type**. **AP** indicates **Anti-Pattern**. **#R** denotes the number of reproduced bugs. **#N** denotes the number of new bugs found. **#TP** and **#FP** indicate the numbers of true positives and false positives, respectively. For anonymization, we omit the bug report links of the two tools; these will be released after paper acceptance.

| Project | BT | AP | Cursor Bugbot | | | | CodeRabbit | | | |
|---|---|---|---|---|---|---|---|---|---|---|
| | | | #R | #N | #TP | #FP | #R | #N | #TP | #FP |
| zstd | OOB | OSO | 0 | 1 | 1 | 0 | 0 | 0 | 0 | 1 |
| systemd | OOB | OSO | 0 | 0 | 0 | 0 | 0 | 0 | 0 | 1 |
| frr | OOB | OSO | 0 | 0 | 0 | 0 | 0 | 0 | 0 | 0 |
| redis | OOB | OSO | 1 | 0 | 1 | 0 | 0 | 0 | 0 | 0 |
| qemu | OOB | OSO | 0 | 0 | 0 | 1 | 0 | 0 | 0 | 1 |
| curl | OOB | NOF | 0 | 0 | 0 | 0 | 0 | 0 | 0 | 0 |
| curl | OOB | NOF | 0 | 0 | 0 | 0 | 0 | 0 | 0 | 0 |
| zstd | OOB | NOF | 0 | 0 | 0 | 0 | 0 | 0 | 0 | 0 |
| php-src | OOB | NOF | 1 | 0 | 1 | 2 | 0 | 0 | 0 | 0 |
| openssl | OOB | NOF | 0 | 0 | 0 | 0 | 0 | 0 | 0 | 0 |
| php-src | OOB | ASO | 0 | 1 | 1 | 0 | 0 | 0 | 0 | 0 |
| systemd | OOB | ASO | 0 | 0 | 0 | 0 | 0 | 0 | 0 | 0 |
| frr | OOB | ASO | 0 | 0 | 0 | 0 | 1 | 0 | 1 | 0 |
| binutils-gdb | OOB | ASO | 0 | 0 | 0 | 0 | 0 | 0 | 0 | 0 |
| gcc | OOB | ASO | 0 | 0 | 0 | 1 | 0 | 0 | 0 | 0 |
| git | DBZ | IZC | 0 | 0 | 0 | 0 | 0 | 0 | 0 | 0 |
| linux | DBZ | IZC | 0 | 0 | 0 | 0 | 0 | 0 | 0 | 0 |
| binutils-gdb | DBZ | IZC | 0 | 0 | 0 | 0 | 0 | 0 | 0 | 0 |
| openssl | DBZ | IZC | 0 | 0 | 0 | 0 | 0 | 0 | 0 | 0 |
| vim | DBZ | IZC | 0 | 0 | 0 | 0 | 0 | 0 | 0 | 0 |
| systemd | DBZ | IZC | 1 | 0 | 1 | 0 | 0 | 0 | 0 | 0 |
| ImageMagick | DBZ | IZC | 0 | 0 | 0 | 0 | 0 | 0 | 0 | 0 |
| ImageMagick | DBZ | IZC | 0 | 0 | 0 | 0 | 0 | 0 | 0 | 0 |
| libuv | DBZ | IZC | 0 | 0 | 0 | 0 | 0 | 0 | 0 | 0 |
| linux | DBZ | LZD | 0 | 0 | 0 | 0 | 0 | 0 | 0 | 0 |
| linux | DBZ | LZD | 1 | 0 | 1 | 0 | 0 | 0 | 0 | 0 |
| linux | DBZ | LZD | 0 | 0 | 0 | 0 | 0 | 0 | 0 | 0 |
| goaccess | DBZ | LZD | 1 | 1 | 2 | 0 | 1 | 1 | 2 | 0 |
| goaccess | DBZ | LZD | 1 | 0 | 1 | 0 | 0 | 0 | 0 | 0 |
| libsass | MLK | UEC | 1 | 0 | 1 | 0 | 1 | 1 | 2 | 0 |
| memcached | MLK | UEC | 0 | 0 | 0 | 2 | 0 | 0 | 0 | 0 |
| memcached | MLK | UEC | 0 | 0 | 0 | 0 | 0 | 0 | 0 | 0 |
| h3 | MLK | UEC | 0 | 0 | 0 | 0 | 0 | 0 | 0 | 0 |
| TrinityEmulator | MLK | UEC | 0 | 0 | 0 | 0 | 0 | 0 | 0 | 0 |
| linux | MLK | MSC | 1 | 0 | 1 | 0 | 1 | 0 | 1 | 0 |
| linux | MLK | MSC | 1 | 0 | 1 | 0 | 1 | 0 | 1 | 0 |
| rtl_433 | MLK | MSC | 0 | 0 | 0 | 0 | 0 | 1 | 1 | 0 |
| libuv | MLK | MSC | 1 | 0 | 1 | 0 | 1 | 0 | 1 | 0 |
| TrinityEmulator | MLK | MSC | 1 | 1 | 2 | 0 | 1 | 0 | 1 | 0 |
| binutils-gdb | MLK | MSC | 0 | 0 | 0 | 0 | 0 | 0 | 0 | 0 |
| **Total** | | | 11 | 4 | 15 | 6 | 7 | 3 | 10 | 3 |

In Listing 2, the function `get_number` sets `val` to 0 if `fgetc` reads the character `'0'` from the input file. This value is then returned and propagated to the function `read_image`, where it is assigned to the variable `maxval`. Subsequently, `maxval` is passed as the second argument to `get_number255`, where it is used as a divisor without any prior validation. While this bug is correctly identified by BUGSCOPE, both BugBot and CodeRabbit fail to detect it. Detecting this issue requires interprocedural data flow tracking between the variables `val` in `get_number` and `maxval` in `get_number255`, along with reasoning about the value range of `val` under different input conditions. Both BugBot and CodeRabbit struggle with such complex, context-dependent vulnerability scenarios.

## H.3 EXAMPLES OF FALSE POSITIVES

In Listing 3, the variable `buf` is assigned the return value of the function `pfile->cb.translate_include`. The function then calculates the length of the con-

Listing 2: A DBZ false negative example missed by Cursor BugBot and CodeRabbit.

```
1 static unsigned int get_number(FILE *fp) {
2   int c, val;
3   c = fgetc(fp);
4   ...
5   val = 0;
6   while (isdigit(c)) {
7     val = 10*val+c-'0';
8     if (is_plain_pbm)
9       break;
10    c = fgetc(fp);
11    if (c == EOF)
12      die("%s: end of file\n", filename);
13  }
14  return val;
15 }
```

```
1 static unsigned int get_number255(FILE *fp, unsigned int maxval) {
2   unsigned int val = get_number(fp);
3   return (255*val+maxval/2)/maxval;
4 }
```

```
1 static void read_image(void) {
2   ...
3   case '2':
4     Plain PGM
5     maxval = get_number(fp);
6     for (i = 0; i < logo_height; i++)
7       for (j = 0; j < logo_width; j++)
8         logo_data[i][j].red = logo_data[i][j].green =
9         logo_data[i][j].blue = get_number255(fp, maxval);
10    break;
11    ...
12 }
```

Listing 3: The OOB false positive example reported by Cursor BugBot.

```
1 bool _cpp_stack_file (cpp_reader *pfile, _cpp_file *file, include_type type,
      location_t loc) {
2   char *buf = nullptr;
3   ...
4   if (!file->header_unit && type < IT_HEADER_HWM
5       && type != IT_INCLUDE_NEXT
6       && pfile->cb.translate_include)
7     buf = (pfile->cb.translate_include(pfile, pfile->line_table, loc, file->path));
8   ...
9   size_t len = strlen (buf);
10  buf[len] = '\n';
11  cpp_buffer *buffer =
12    cpp_push_buffer (pfile, reinterpret_cast<unsigned char *> (buf), len, true);
13  buffer->to_free = buffer->buf;
14  ...
15 }
```

tent in `buf` using `strlen(buf)`, and sets the next position to `'\n'`. BugBot reports a buffer overflow for this operation. However, `strlen(buf)` only describes the length of valid characters in `buf` up to the null terminator, and does not necessarily reflect the actual allocated size of the buffer. Without enough context regarding the allocation of `buf`, we cannot conclusively determine that `strlen(buf)` represents the actual length of the buffer.

In Listing 4, the variable `buf` is allocated a fixed memory region of size `LIST_SIZE_INCREASE`, and `bufend` is set to point to the end of this buffer. In the loop, each element of `fileNamesTable` is assigned an offset within `buf` using `fileNamesTable[ifnNb] = buf + pos`. Before accessing the memory, the program checks `if (buf + pos > bufend)` to ensure the access stays within bounds. CodeRabbit flags this comparison as a potential undefined behavior due to possible pointer overflow in `buf + pos`. However, `pos` is incremented gradually based on the length of the input filenames using `pos += strlen(fileNamesTable[ifnNb]) + 1`. Triggering an overflow in `buf + pos` would require `pos` to approach the maximum value of `size_t`, which in practice would require an unrealistically large number of large input strings (e.g., more than $10^{18}$). Therefore, this report is a false positive, as such extreme conditions are virtually impossible to reach in realistic scenarios.

Listing 4: The OOB false positive example reported by CodeRabbit.

```
1  FileNamesTable* UTIL_createExpandedFNT(const char* const* inputNames, size_t nbIfns,
     int followLinks) {
2   unsigned nbFiles;
3   char* buf = (char*)malloc(LIST_SIZE_INCREASE);
4   char* bufend = buf + LIST_SIZE_INCREASE;
5   ...
6   size_t ifnNb, pos;
7   for (ifnNb = 0, pos = 0; ifnNb < nbFiles; ifnNb++) {
8     fileNamesTable[ifnNb] = buf + pos;
9     if (buf + pos > bufend) { free(buf); free((void*)fileNamesTable); return NULL; }
10    pos += strlen(fileNamesTable[ifnNb]) + 1;
11  }
12  ...
13 }
```

